# Characterization and Bioactivity of *Piper chaudocanum* L. Extract-Doped ZnO Nanoparticles Biosynthesized by Co-Precipitation Method

**DOI:** 10.3390/ma16155457

**Published:** 2023-08-03

**Authors:** Thi Thao Truong, Thi Tam Khieu, Huu Nguyen Luu, Hai Bang Truong, Van Khien Nguyen, Truong Xuan Vuong, Thi Kim Ngan Tran

**Affiliations:** 1Faculty of Chemistry, TNU-University of Sciences, Tan Thinh Ward, Thai Nguyen City 250000, Vietnamkhiennv@tnus.edu.vn (V.K.N.); xuanvt@tnus.edu.vn (T.X.V.); 2Laboratory of Magnetism and Magnetic Materials, Science and Technology Advanced Institute, Van Lang University, Ho Chi Minh City 700000, Vietnam; luuhuunguyen@vlu.edu.vn; 3Faculty of Applied Technology, School of Technology, Van Lang University, Ho Chi Minh City 700000, Vietnam; truonghaibang@vlu.edu.vn; 4Optical Materials Research Group, Science and Technology Advanced Institute, Van Lang University, Ho Chi Minh City 700000, Vietnam; 5Institute of Applied Technology and Sustainable Development, Nguyen Tat Thanh University, Ho Chi Minh City 700000, Vietnam; 6Faculty of Environmental and Food Engineering, Nguyen Tat Thanh University, Ho Chi Minh City 700000, Vietnam

**Keywords:** green synthesis, ZnO nanoparticles, *Piper chaudocanum* L., antibacterial, anticancer

## Abstract

Green synthesis and nanomaterials have been the current trends in biomedical materials. In this study, *Piper chaudocanum* L. leaf extract-doped ZnO nanoparticles (PLE-doped ZnO NPs), a novel nanomaterial, were studied including the synthesis process, and the biomedical activity was evaluated. PLE-doped ZnO NPs were synthesized by the co-precipitation method, with differences in the synthesis procedures and dosages of the extract. The X-ray diffraction, Fourier transform infrared, scanning electron microscopy, energy dispersive X-ray spectroscopy, Brunauer–Emmett–Teller, ultraviolet-visible diffuse reflectance spectroscopy, and photoluminescence spectrum analysis results showed that the biosynthesized PLE-doped ZnO NPs were pure and in a hexagonal wurtzite phase. The PLE-doped NPs were synthesized by adding the extract to the zinc acetate solution before adjusting the pH and exhibited the smallest size (ZPS50 was 22 nm), the richest in the surface organic functional groups and the best optical activity. The highest antibacterial activity against *P. aeruginosa* and *S. aureus* was observed at 100 µg/mL of ZPS50 NPs, and the inhibition zone reached 42 and 39 nm, respectively. Moreover, ZPS50 NPs showed a moderate effectiveness against KB cancer cells with an IC_50_ value of 43.53 ± 2.98 µg/mL. This present study’s results suggested that ZPS50 NPs could be a promising nanomaterial in developing drugs for treating human epithelial carcinoma cells and infectious illnesses.

## 1. Introduction

Nanomaterials and nanotechnology offer a significant advantage in science and technology, creating many useful products serving many areas of life, including medicine [1]. Among them, ZnO NPs have emerged as one of the most extensively studied oxides across multiple domains (as a multifunctional material): electronics, optoelectronics, piezoelectric, spintronic, and laser technology; sensors, converters, and energy generators; the rubber industry and ceramic industry; absorbents, catalysts, and photocatalysis; and food, sunscreen lotion, or other biological applications and biomedical applications [2,3,4]. The worldwide production of ZnO in 2020 was estimated at 45–46 thousand tons, out of a total of approximately 660 thousand tons of metal oxides [5]. They were widely studied due to their intriguing properties, including non-toxicity, chemical stability, biocompatibility, biodegradability in nature, and low cost [6]. Recently, compared with other metal oxide nanomaterials, ZnO NPs have achieved much exciting progress in biomedicine applications, such as antibacterial, anti-inflammation, wound healing, anticancer, drug delivery, and bioimaging [7].

Many physical and chemical techniques, such as sol–gel, hydrothermal, mechanochemical, hydrometallurgy, and thermal decomposition, have been employed to synthesize ZnO NPs [8]. Compared to these techniques, the synthesis of ZnO NPs by the green chemistry approach has gained considerable momentum [9]. This method uses renewable, non-toxic, and ecologically benign materials, such as microbes, enzymes, and natural extracts, while consuming minimal energy and being inexpensive [10,11].

Plant extracts, which are readily available and inexpensive, contain various compounds, such as nucleic acids, carbohydrates, alkaloids, terpenoids, flavonoids, and steroids; different parts of plants, including leaves, roots, rhizomes, barks, fruits, and flowers have been used to synthesize ZnO NPs [12]. These biological substances are natural and biodegradable, making them relatively safer than hazardous chemicals. Furthermore, plants are the primary source of traditional herbal medicine, and doping plant extracts on materials has been observed to enhance the materials’ antibacterial, cytotoxic, or antioxidant abilities more than those synthesized through chemical methods. For instance, ZnO NPs were synthesized using *Pandanus odorifer* leaf extract, which inhibited the growth of the MCF-7, HepG2, and A-549 cells at different concentrations ranging from 50 to 100 μg/mL, along with antibacterial activity against *B. subtilis* (26 nm) and *E. coli* (24 nm) at 50 μg/well [13]. The ZnO NPs were synthesized using *Tecoma castanifolia* leaf extract, and this performed excellent inhibitory activity against *B. subtilis*, *S. aureus*, *E. coli*, and *P. aeruginosa* and significant anticancer activity against A549 cells with an IC_50_ value of 65 μg/mL [14]. Similarly, the biosynthesis of ZnO NPs using extracts of *Swertia chirayita* [15] or *Costus pictus* [16] exhibited various antimicrobial and anticancer activities. Therefore, biosynthesized ZnO NPs have been considered a bio-safe material and pivotal in several bacterial and cancer research studies [14,17]. The plant extracts act as a green solvent, a green precursor (a complexing agent, a reducing agent, an oxidizing agent, an acid or base agent, etc.), or surfactant (stabilizing agent) for the reaction or work together with Zn (II) ions and facilitate the formation of ZnO NPs [8,18].

The *Piper (P.)* genus has been traditionally used to treat various ailments, including headaches, liver ailments, body aches, epilepsy in children, colds, flu, and as an antidote for various toxins and weather-related illnesses [19]. Several *Piper* species, such as *P. longum* [20], *P. betle* [21], and *P. nigrum* [22] have been employed as reducing and capping agents in the production of ZnO NPs. Recently, *P. chaudocanum* L., a member of the *Piper* genus, has been in employment to synthesize nanoparticles [23,24,25,26] because it comprises various compounds such as alkaloids, triterpenoids, flavonoids, steroids, tannins, phenolic compounds, and piperine. This amide alkaloid is considered to be a chief constituent [23]. For instance, the extract of *P. chaudocanum* L. stems and essential oils from *P. chaudocanum* leaves were used as a reducing agent and stabilizing agent for the biosynthesis of AgNPs from AgNO_3_ [23,24]. The prepared AgNPs showed a high antibacterial activity against ailments, and they were an Hg^2+^ colorimetric detection in a solution. The extract of *P. chaudocanum* L. leaf was used as a green precursor in the biosynthesis of ZnO NPs [25] and Cu-doped ZnO NPs [26] by the sol–gel method, replacing conventional precursors such as citric acid or tartaric acid; the prepared ZnO NPs and Cu-doped ZnO NPs showed an effective adsorption of Pb^2+^ and Cu^2+^, respectively.

In this study, *P. chaudocanum* L. leaf extract-doped ZnO nanoparticles (PLE-doped ZnO NPs) were biosynthesized by a co-precipitation method at room temperature as a novel nanomaterial; the effect of the synthesis process sequence and the extract dosages on the characteristics and applications for biomedicine have been investigated. The *P. chaudocanum* L. leaf extract acts as a surfactant and an additional precursor to modify the material surface, improving the prepared nanomaterials’ performance. Interestingly, our results found the best order of participation of *P. chaudocanum* L. extracts in the fabrication of ZnO NPs by the co-precipitation method, which enhances the biomedical application of PLE-doped ZnO NPs.

## 2. Materials and Methods

### 2.1. Materials

Analytical chemicals, including NaOH (98%), Zn(CH_3_COO)_2_·2H_2_O (98%), and C_2_H_5_OH (99%) were purchased from Sigma-Aldrich Chemicals (Merck KGaA, Darmstadt, Germany). *P. aeruginosa* and *S. aureus* were isolated and cultured in the Biochemical Department, Chemical Institute, Vietnam Academy of Science and Technology.

### 2.2. Preparation of Piper chaudocanum Leaf Extract

*P. chaudocanum* L. leaves were collected in Vietnam and then thoroughly rinsed with distilled water to remove contaminants. Subsequently, they were air-dried at 50 °C and ground using a grinder to obtain a fine powder. The extraction of *P. chaudocanum* L. leaves was performed, following the procedure outlined in our previous study [24]: 10 g of *P. chaudocanum* L. leaves powder was extracted via ultrasound, by a 100 mL mix of ethanol and distilled water (volume ratio 1:1) at 60 °C for 1 h, then cooled, centrifugated, filtered to collect the extract, and noted as *P. chaudocanum* extract and preserved at 4 °C for the synthesis of PLE-doped ZnO NPs in subsequent experiments.

### 2.3. Synthesis of PLE-Doped ZnO NPs

This present study involved the synthesis of PLE-doped ZnO NPs via the co-precipitation method, employing different sequences of *P. chaudocanum* extract in addition to the zinc acetate solution.

In the first process, PLE-doped ZnO NPs were synthesized following the improved procedure described by S. Vasantharaj et al. [27]. First, 25, 50, and 75 mL of the *P. chaudocanum* L. extract were separately added to 50 mL of a 0.1 mM aqueous zinc acetate solution and stirred for 60 min at 70 °C. This immediately changed the color from colorless to a pale yellow in the solution. After that, the pH of the solution was adjusted by adding 1.0 M NaOH solution under constant stirring until the pH reached 10. The mixture was then stirred for 60 min at 70 °C. The formed pale-yellow precipitates were collected by centrifugation at 5000 rpm for 10 min, washed with distilled water to achieve a neutral pH, dried, and characterized. The PLE-doped ZnO NPs samples obtained from this process were labeled ZPS25, ZPS50, and ZPS75.

In the second process (following the improved procedure described by M. Shabaani et al. [28]) the following occurred: a similar process was repeated; however, the *P. chaudocanum* L. extract was added to the reaction mixture after the reaction between the zinc acetate solution and NaOH solution. The resulting PLE-doped ZnO NPs were labeled ZSP25, ZSP50, and ZSP75.

Moreover, the ZnO NPs were additionally synthesized chemically through a similar process however without adding the *P. chaudocanum* L. extract. The resulting product was labeled as CHE-ZnO NPs, without the addition of the *P. chaudocanum* L. extract. The resulting product was labeled as CHE-ZnO NPs.

### 2.4. Characterizations of the Synthesized Materials

Characterizations of the ZnO NPs samples were recorded by using various techniques, including X-ray diffraction (XRD) (D2-Phaser, Brucker, Karlsruhe, Germany) with 2 theta from 20° to 80°; Fourier transform infrared spectroscopy (FTIR) (Spectrum Two, PerkinElmer, Shelton, CT, USA) at a wavenumber band of 4000–400 cm^−1^; scanning electron microscopy (SEM) (Leo 1430VP, Carl Zeiss AG, Jena, Germany) [15] and field emission scanning electron microscopy (FE–SEM) (Regulus 8100, Hitachi Hight-Tech Corporation, Tokyo, Japan); integrated, energy dispersive X-ray spectroscopy (EDS), using the Bruker Xflash 6130 probe (Bruker, Billerica, MA, USA); Brunauer–Emmett–Teller (BET) surface area analysis (TriStar 3000 V6.07, Micromeritics Instrument Corporation, Norcross, GA, USA); ultraviolet-visible diffuse reflectance spectroscopy (UV-Vis-DRS) (Scinco 4100, Shimadzu, Kyoto, Japan), in the wavelength ranging from 250 nm to 700 nm, step 0.5 nm using cm^−1^ quartz cuvettes, step 0.5 nm [22]; and photoluminescence (PL) (FLS 1000, Edinburgh Instruments Ltd., Livingston, West Lothian, UK) within the wavelength ranges of 375 nm to 800 nm (step 1 nm) were obtained under the 325 nm line of Xe laser excitation [29].

### 2.5. Determination of the Antibacterial Activity of PLE-Doped ZnO NPs

The antibacterial activity of the synthesized PLE-doped ZnO NPs was evaluated by the agar diffusion method, as described by [30], against two bacteria strains, namely, *P. aeruginosa* (Gram-negative) and *S. aureus* (Gram-positive). The bacterial cultures were prepared by suspending the strains at 4–5 × 10^8^ CFU/mL concentrations and spread on Luria–Bertani agar plates. A total of 4 wells with diameters of 8 mm were produced on these plates. Subsequently, 50 µL of each sample at a 100 μg/mL concentration was placed into each well at aseptic conditions, and a 2% DMSO solution as a negative control was poured into the separated wells. These plates were then incubated for 24 h at 37 °C, before measuring the inhibition zones.

### 2.6. Determination of the Anticancer Activity of the ZPS50 Sample

The cytotoxic activity of the ZPS50 sample against the KB human cancer cell line, acquired from the American Type Culture Collection, was evaluated using a 3-(4,5-dimethylthiazol-2-yl)-2,5-diphenyltetrazolium bromide (MTT) assay [31]. The KB cells were cultured in Dulbecco’s Modified Eagle’s Medium, supplemented with 10% fetal bovine serum, 1% L-glutamine, penicillin, and streptomycin at 37 °C in a humidified 5% CO_2_ atmosphere. The MTT assays were performed by splitting the cells using trypsin (3 × 10^4^ cells/mL), which were then seeded in 96-well plates before treatment with a series of concentrations (4.0, 16.0, 64.0, and 256.0 µg/mL) of the samples dissolved in DMSO (20 mg/mL). For the MTT assay, untreated cells were used as controls. After 72 h of treatment, 10 µL of a 5 mg/mL MTT solution in a phosphate buffer was added to each well, and then the cells were incubated for 4 h. After removing the MTT, 100 μL of DMSO solution was added to each well to dissolve the formazan crystals. The solution’s optical density (OD) was measured at 540 nm, using a plate reader (BIOTEK). All experiments were carried out in triplicate, and ellipticine was used as a positive control. The cell viability was calculated by Equation (1), and the inhibition curve was used to calculate the IC_50_ values graphically.
(1)$$\%\;cell\;viability=\frac{1-{OD}_{sample}}{{OD}_{control}}\times 100\%$$
where OD_sample_ and OD_control_ reflect the optical densities of the samples and control.

## 3. Results

### 3.1. Characteristics of Synthesized ZnO NPs

#### 3.1.1. Phase Structure and Atom Composition of Synthesized ZnO Samples

The synthesized samples’ crystalline nature and phase structure were analyzed using X-ray diffraction (XRD) over the 2θ range from 20° to 80° and are shown in Figure 1a. The atom composition of some samples was determined by the EDS technique and shown in Figure 1b.

In Figure 1, the XRD patterns of all the samples, except for ZPS75, displayed 9 diffraction peaks at 2θ values of 31.6°, 34.2°, 36.0°, 47.3°, 56.4°, 62.7°, 66.3°, 67.8°, and 69.0°, corresponding to the Miller plane (1 0 0), (0 0 2), (1 0 1), (1 0 0), (1 1 0), (1 0 3), (2 0 0), (1 1 2), and (2 0 1), respectively, and match well with the PDF card (JCPDS No.36-1451 [18]) of the powder diffraction standards of ZnO, which confirmed the purity and hexagonal wurtzite phase of PLE-doped ZnO NPs and CHE-ZnO NPs [32]. The ZPS75 spectrum showed a high degree of complexity, characterized by the presence of ZnO diffraction peaks as well as numerous diffraction peaks arising from other impurities. Upon a comparison of the XRD patterns of all the samples, the diffraction peaks of ZPS and ZSP samples shifted slightly towards smaller 2θ values relative to those of the CHE-ZnO NPs samples.

Particle size determination was carried out for CHE-ZnO NPs, ZPS25, ZPS50, ZPS75, ZSP25, ZSP50, and ZSP75 using the Scherrer equation, yielding average crystallite sizes of 35, 32, 22, 27, 40, 33, and 30 nm, respectively. The particle sizes of ZPS were smaller than those of ZSP, indicating that the order of the reaction steps played a crucial role in determining crystallite sizes. It is known that the introduction of an alkaline solution into a Zn (II) solution triggers a chemical reaction:(2)$${\mathrm{Zn}}^{2+}+2{\mathrm{OH}}^{-}\;\to \;{\mathrm{Zn}(\mathrm{OH})}_{2}$$
(3)$$\mathrm{Zn}(\mathrm{OH}{)}_{2}\;\overset{t^{o}}{\to }\;\mathrm{ZnO}\cdot H_{2}O$$

When the *P. chaudocanum* L. extract was introduced into the Zn (II) solution after the addition of the alkaline hydroxide solution, the extract would interact with ZnO·H_2_O as a surfactant and adsorb on the ZnO·H_2_O surface, therefore, modifying the initial surface of ZnO·H_2_O without affecting the formation of ZnO·H_2_O. However, in the case that the *P. chaudocanum* L. extract was added into the Zn (II) solution before the addition of the alkaline hydroxide solution, an interaction between the Zn (II) ions and the *P. chaudocanum* L. extract would occur, which was reflected by a reduction in the XRD diffraction peak intensity and an increase in the diffraction peak width of ZPS samples versus ZSP samples. The research [23] reported that the *P. chaudocanum* L. extract contained a wide range of substances, including alkaloids, triterpenoids, flavonoids, steroids, tannins, phenolic compounds, and piperine, an amide alkaloid that was considered to be a key component [23]. This suggested the formation of a complex between Zn (II) ions and the *P. chaudocanum* L. in the case of the ZPS samples, which are explained by the Equations (4) and (5) below:(4)$${\mathrm{Zn}}^{2+}+P.\;\mathrm{chaudocanum}L.\;\mathrm{extract}\;\to \;{\mathrm{Zn}}^{2+}.P.\;\mathrm{chaudocanum}L.\;\mathrm{extract}$$
(5)$${\mathrm{Zn}}^{2+}.\;P.\;\mathrm{chaudocanum}\;L.\;\mathrm{extract}+2{\mathrm{OH}}^{-}\;\to \;\mathrm{ZnO}.H_{2}O.\;P.\;\mathrm{chaudocanum}\;L.\;\mathrm{extract}$$

In our previous study, ZnO NPs were green synthesized by the sol–gel method using *P. chaudocanum* L. extract [23,26]; the extract acted as a precursor (a complexing agent) following Equation (4), and then, the complex was burned at approximately 300 °C to 500 °C to produce ZnO:(6)$${\mathrm{Zn}}^{2+}.\;P.\;\mathrm{chaudocanum}\;L.\;\mathrm{extract}+O_{2}\;\to \;\mathrm{ZnO}\;+\;{\mathrm{CO}}_{2}\;+\;H_{2}O\;+\;\mathrm{other products}$$
and the smallest ZnO NPs were 30.6 nm [27]. However, in this study, the smallest ZnO was 22 nm (ZPS50). This result showed that the different roles of *P. chaudocanum* L. extract in the synthesis process significantly affect the formation of ZnO crystal, such as the particle size and possibly surface characteristics, which led to the enhancement of the biomedical applicability of the ZnO NP. Furthermore, the increasing volume of the *P. chaudocanum* L. extract showed a noticeable decrease in crystallite size. This finding suggested that the *P. chaudocanum* L. extract volume significantly influenced the formation of the resulting products, particularly the ZPS samples. In the role of a surfactant, the 25 mL extract did not reduce however additionally increased the size of the ZnO particles. As the volume of the extract increased from 50 mL to 75 mL, the particle size decreased gradually (33 and 30 nm) compared to the absence of extract (35 nm), demonstrating the effective activity of the extract. In the role of a complexing agent, as the extract volume increased from 25 to 50 mL, the particle size significantly reduced from 32 down to 22 nm; however, at 75 nm, the obtained product was contaminated with impurities. It showed that the larger quantities of organic compounds in the extract reacted with Zn^2+^ to form undesirable by-products. Therefore, 50 mL of the extract was the most appropriate volume.

The element composition of some samples was determined by the EDS technique (Figure 1b), confirming the presence of organic compounds in the synthesized ZnO NPs. The CHE-ZnO sample only consisted of Zn and O; however, in all the PLE-doped ZnO NPs samples C and N were present in different concentrations. Depending on the different synthesis processes, the ratio of C and N in the product was different: the C content in the product which was synthesized following the first process (ZPS) was higher than those synthesized by the second process (ZSP); however, the N content was in reverse. It means that, in the different reaction order, the various organic compounds (in the PC extract) interacted with different reactants (ZnO·H_2_O or Zn^2+^) or bi-modified during reactions 5 and 6 and attacked the ZnO structure. It led to the formation of ZPS and ZSO, which differed in surface organic composition and properties (bioactive). The common point is that as the amount of used extract increased, the content of C and N in the product increased.

#### 3.1.2. FTIR Spectra of PLE-Doped ZnO NPs and CHE-ZnO NPs

The potential interactions between biomolecules and the surfaces of ZnO nanoparticles were investigated by using Fourier transform infrared spectroscopy (FTIR), at a wave number band of 4000–400 cm^−1^, as shown in Figure 2.

The FTIR spectra indicated that all the samples displayed a prominent absorption peak at approximately 420 cm^−1^, corresponding to the vibration of the Zn-O bond [33]. The FTIR spectra of the ZPS and ZSP samples exhibited characteristic absorption peaks for the vibrations of bonds in organic compounds, whereas the CHE-ZnO NPs sample lacked these vibrations. Notably, all ZPS and ZSP samples showed a broad absorption peak in the 3200–3600 cm^−1^ range, which could be attributed to the stretching vibration of the O-H bond in phenolic or flavonoid and carboxylic acid compounds [34]. Additionally, the absorption peak observed at 2800–2900 cm^−1^ was attributed to the symmetric and asymmetric vibrations of the C-H bonds presented in all organic compounds [35]. The outcomes demonstrated that the organic compounds in the *P. chaudocanum* extract participated in the formation of ZnO NPs and acted as capping agents on the surface of ZnO NPs to prevent their agglomeration. In addition, a peak observed at a wavenumber of 550 cm^−1^ was assigned to the E2 form of the hexagonal ZnO wurtzite crystal lattice structure [36], indicating that the presence of *P. chaudocanum* L. extracts in the synthesis process affected the formation of the crystal lattice structure of the ZnO NPs. When comparing the FTIR spectra of the ZPS and ZSP samples, the strong absorption peak at 1680 cm^−1^ in the FTIR spectrum of the ZPS samples was typical for the stretching vibration of the C=O bond, which was presented in the extract and formed during the reduction process. In contrast, the FTIR absorption peak in the region of the ZSP samples was weak.

This result proved that the synthesis process of ZPS samples was more optimal than that of ZSP samples. Furthermore, the FTIR spectra of the ZPS samples showed an absorption peak in the region of 2220–2500 cm^−1^, corresponding to the stretching vibration of the C≡N bond. Moreover, the FTIR spectrum of ZPS50 showed an absorption band at 1100–1200 cm^−1^, which could be attributed to the C-O stretching vibration and absorption peaks at 1340 cm^−1^ and 1480 cm^−1^, which could be associated with C-N and C=C stretching vibrations, respectively. This result was suitable to the result reported by the previous study [23]. The addition of the *P. chaudocanum* L. extract into the Zn (II) solution before the addition of the sodium hydroxide solution in the synthesis process led to the formation of organic functional groups on the ZnO surface, whereas the process in which the *P. chaudocanum* L. extract was added into the Zn (II) solution after the addition of the sodium hydroxide solution showed this effect noticeably weaker. Moreover, 50 mL of extract was more suitable than 25 mL. This finding was consistent with the results of XRD.

#### 3.1.3. Surface Characterizations

The scanning electron microscope (SEM) was utilized to investigate the microstructure and morphology of the samples, as depicted in Figure 3.

The SEM images revealed that the CHE-ZnO NPs exhibited a clumped, tree-like block structure, and the particle size was approximately 25 nm (Figure 3a). The particles of the ZSP samples were thin sheets, with ZSP25 sheets being thicker and larger than those of ZSP50 (Figure 3b,c). The particle sizes of ZSP25 and ZSP50 NPs were approximately 18 and 16 nm. The ZSP50 particles adhered tightly, and it was difficult to observe the grain boundaries. However, the morphology of ZPS25 was similar to that of the CHE-ZnO NPs, albeit more porous (Figure 3d). Particles (approximately 12–14 nm) stick together into relatively thick, leaf-shape spongy masses, approximately 40–50 nm wide and 100–200 nm long. The ZPS50 particles (approximately 10–12 nm; see Figure 3e) form a layered structure resembling a multi-petal flower, with each petal resembling a spiky, oblong leaf, the widest part being approximately 50 nm and 70–140 nm long; the leaves were attached to each other and spread out in 3 directions, such as a pine tree (Figure 3f). This structure is called a nano-flower, which usually has a rough surface and high porosity, an increasing orientation effect in specific directions, and good bioactivity [37,38]. The estimated particle size from the SEM images is consistent with the estimated particle size variation trend calculated from the XRD data using the Scherrer equation, however with a smaller value.

The nitrogen absorption–desorption experiments of the CHE-ZnO and ZPS50 NPs are recorded in Figure 4. The linear isotherm plot was the intermediate form between type II and III [39]; the mean pore diameters of CHE-ZnO and ZPS50 NPs were 47.2652 nm and 33.7243 nm, respectively, which indicated both CHE-ZnO and ZPS50 NPs samples were the mesoporous structure. The uptake of N_2_ was initially slow, and until the surface coverage was sufficient, the interactions between adsorbed and free molecules began to dominate the process. The BET surface areas of the CHE-ZnO and ZPS50 NPs samples were 8.3587 m^2^/g and 10.8486 m^2^/g, respectively. This result confirms the reduction in grain size of ZPS50, compared with the CHE-ZnO particles.

#### 3.1.4. The Optical Properties of Prepared Samples

The optical properties of the synthesized ZnO NPs were analyzed through the UV-VIS DRS technique, and the results are illustrated in Figure 5. The UV-Vis DRS spectrum of the CHE-ZnO NPs manifested a strong absorption peak at 366.5 nm, with minimal absorbance in the visible region (see inset of Figure 5). All the PLE-doped ZnO NPs samples had a greater intensity of absorbance than the CHE-ZnO NPs. Compared to the CHE-ZnO NPs, the ZPS samples demonstrated a slight red-shift of the highest absorption peak from 366.5 to 370.5 nm. This trend was additionally found in the previous works [25,40,41]. The absorbance intensities of the ZSP samples were higher than those of the ZPS samples, and the absorption peaks were shifted to a smaller wavelength, approximately 344.5 to 347.0 nm. This observation suggested that the reaction order of the Zn^2+^ solution influenced the structure and surface properties of the synthesized PLE-doped ZnO NPs, *P. chaudocanum* L. extract, and alkaline solution, which altered the optical properties of the materials. The high absorbance intensity was good; however, the absorption peak of the ZSP NPs at smaller wavelengths (344.5 to 347 cm^−1^, compared to 366.5 cm^−1^ of the CHE-ZnO NPs) additionally meant a larger energy band gap, and they were more challenging to activate in the region of visible light. In contrast, the absorbance intensities of the ZPS NPs were smaller, and the absorption peak shifted to a larger wavelength (370 cm^−1^), decreasing the energy band gap. In addition, the absorbance intensities of the ZPS NPs were smaller than the ZSP NPs however much higher than the CHE-ZnO NPs. As a result, the ZPS NPs would be more active and beneficial for bioactivity in the region of visible light [42].

Additionally, the PL emission spectra (excitation at 325 nm) were acquired and are presented in Figure 6.

The PL spectra of the synthesized samples were analyzed, revealing emission peaks at 387 and 405.5 nm. The peak at 387 nm was assigned to near-band edge emission (NBEM), which resulted from the radiative recombination of the free excitons from the exciton–exciton collision process [43] or free electron-hole recombination [44]. The emission peaks were closely associated with the ZnO band gap [45], showing a weak UV emission. The peak at 405.5 nm reflected the existence of defect levels in ZnO. Specifically, the non-radiative Zn interstitial defects would be initiated either by the transition from the Zn_i_ to the valence band position or from the conduction band to the O_i_ level [46]. The intensity of these peaks in the presence of the *P. chaudocanum* L. extract decreased in the following order: CHE-ZnO NPs > ZPS25 > ZSP50 > ZPS50 ~ ZPS25. All the samples exhibited minor peaks between 430 and 500 nm in the visible PL region. The PL spectrum of the CHE-ZnO NPs showed a broad and intense peak from 425 nm to 700 nm, which can be attributed to shallow and deep traps and oxygen defects [46]. The observed broad peak in the visible region was attributed to impurity levels, correlated with the singly ionized oxygen vacancy in ZnO and/or the transition between photo-excited holes [47]. The intensity of this peak significantly decreased in the PLE-doped ZnO NPs samples, following the order of CHE-ZnO NPs >> ZPS25 > ZPS50 > ZSP25 > ZSP50. Crystalline defects, such as zinc interstitials and oxygen vacancies, explained the appearance of the visible region peak [48]. The presence of defects, including impurity atoms occupying substitutional or interstitial lattice sites, vacancies, or self-interstitial, would considerably alter the luminescent properties of the synthesized samples. The remarkable reduction in visible PL intensity reveals the defects at the ZnO NPs’ surfaces [43] and the role of the *P. chaudocanum* L. extract in improving the charge separation [46] and diminishing oxygen (vacancy) defects in the ZnO lattice structure [49]. However, the result was not significant for the ZSP sample due to its poor activity in visible light (UV-Vis DRS results); therefore, the ZPS50 samples showed the best optical activity.

Overall, the results of the XRD pattern, EDS pattern, FTIR spectra, SEM images, BET, UV-Vis DRS, and PL spectra demonstrated the following: the PLE-doped ZnO NPs samples which were synthesized following the first process (the Zn^2+^ ion interacted with the *P. chaudocanum* L. extract, before adjusting the pH by the addition of the alkaline solution) were smaller in particle size, richer in organic functional groups on the ZnO surface, more porous, with more optical activity, and more active in the visible light than the PLE-doped ZnO NPs samples which were synthesized following the reverse process. The most suitable extract volume was 50 mL, and the ZPS50 sample was expected to have the best applicability.

### 3.2. Antibacterial Activity

The synthesized samples were evaluated as antibacterial against two pathogenic bacteria strains, *P. aeruginosa* (Gram-negative) and *S. aureus* (Gram-positive), by the agar diffusion method. The result is performed in Figure 7 and Appendix A. All the samples showed an effective inhibition against both pathogens, with the inhibition zones at a 100 μg/mL concentration ranging from 25 to 42 mm for *P. aeruginosa* and 22 to 39 mm for the *S. aureus* bacterial. Notably, the samples with smaller crystallite sizes displayed more potent antibacterial activity than those with larger ones, and the antibacterial activity of the ZPS samples was higher than the ZSP samples. The ZPS50 sample, with the smallest particle size, exhibited the strongest antibacterial activity against *P. aeruginosa* and *S. aureus*, with inhibition zones of 42 and 39 mm, respectively.

Conversely, the CHE-ZnO NPs with the largest crystallite size displayed the weakest activity. A decrease in particle size (from 35 to 22 nm) explicitly increased the diameters of the inhibition zones of all bacteria. This phenomenon could be attributed to the smaller particles’ higher surface area and volume ratio, which enabled more efficient antibacterial activity [50]. Despite a slight difference in particle sizes between the CHE-ZnO NPs (35 nm) and ZPS25 (32 nm), their antibacterial efficacy significantly increased from 25 to 31 mm for *P. aeruginosa* and 22 to 35 mm for *S. aureus* bacteria, respectively. This observation indicated that the capping agents, such as piperine, beta-sitosterol, and campesterol glucoside, in the *P. chaudocanum* L. extract, were attached to the nanoparticle surface, which could adhere to bacterial membranes and enter and inhibit the bacteria [50,51]. These values were better than the antibacterial activity of some of the published biosynthetic ZnO NPs, such as *P. nigrum* (27.6 mm inhibition zone diameter for *S. aureus*) [22], *P. betle* (3 mm for *S. aureus*) [52], *Becium grandiflorum* (7 mm for *S. aureus*, and 11 mm for *P. aeruginosa bacteria*) [53], and *Arthrospira platensis* (21.1 mm for *S. aureus*, and 19.1 mm for *P. aeruginosa*) [54]. This result indicated that the PLE-doped ZnO NPs in this study had more potential as antibacterial drugs for bio-related applications. The results were in agreement with the previous studies, which reported that the antibacterial capability of nanoparticles depends highly on the physicochemical properties of the nanoparticles, including stability, size, shape, and surface charge, as well as capping agents [55,56,57]. The ZPS50 NPs exhibited the best antibacterial activity for 2 reasons: (1) their diameters were the smallest, and (2) their surface was the richest in organic functional groups from the *P. chaudocanum* L. extract, to which they were attached.

### 3.3. Anticancer Activity of ZPS50 Sample

The KB cancer cells were treated with different doses of ZPS50 NPs (4–256 μg/mL) for 72 h, and the results are presented in Appendix A. Ellipticine was utilized as the positive control (IC_50_ = 0.31 ± 0.05 µg/mL). The results indicated a significant increase in the cancer cells’ inhibition percentage from 0 to 81%, when the ZPS50 concentration was increased from 4 to 256 µg/mL. As shown in Appendix A, the ZPS50 sample showed moderate effectiveness against KB cells with the IC_50_ value of 43.53 ± 2.98 µg/mL. The anticancer activity of the ZPS50 sample was comparable to that of ZnO synthesized by a chemical method (IC_50_ value of 38.4 μg/mL) in another study [58]. Notwithstanding, ZPS50 was more effective against KB cells than AgNPs synthesized using *Helicteres isora* stem bark extract [59] and *Argyreia nervosa* leaf extract [60] with IC_50_ values of 58.64 μg/mL, respectively. These findings suggest that ZPS50 NPs, PLE-doped ZnO NPs could be a promising material for developing drugs for treating human epithelial carcinoma cells.

## 4. Conclusions

Biochemical *P. chaudocanum* L. extract-doped ZnO NPs (PLE-doped ZnO NPs) were successfully synthesized through conventional co-precipitation. XRD, EDS, FTIR, PL, UV-VIS DRS, SEM, and BET analyses showed that the synthesis process sequence and the additional dosages of *P. chaudocanum* L. extract significantly impacted the morphology, particle size, surface, and optical property of the materials. The reaction between the Zn (II) ion and *P. chaudocanum* L. extract before adding an alkaline solution formed the smallest nanoparticles, with the most intense organic functional groups on their surface. The smallest particle size was observed in the ZPS50 NPs (~22 nm), using 50 mL of *P. chaudocanum* L. extract. Furthermore, the antimicrobial activity of the PLE-doped ZnO NPs was assessed against Gram-positive (*P. aeruginosa*) and Gram-negative (*S. aureus*) bacteria, which exhibited an improved efficacy compared to chemically synthesized CHE-ZnO NPs. Specifically, the largest inhibition zone was observed at 100 µg/mL of ZPS50, reaching 42 and 39 nm against *P. aeruginosa* and *S. aureus* bacteria, respectively. The cytotoxic activity of the ZPS50 NPs against the KB human cancer cell line was determined with the IC_50_ value of 43.53 ± 2.98 µg/mL. These findings suggest that ZPS50 NPs and PLE-doped ZnO NPs could be potential materials for developing drugs for treating human epithelial carcinoma cells and infectious illnesses.

## Figures and Tables

**Figure 1 materials-16-05457-f001:**
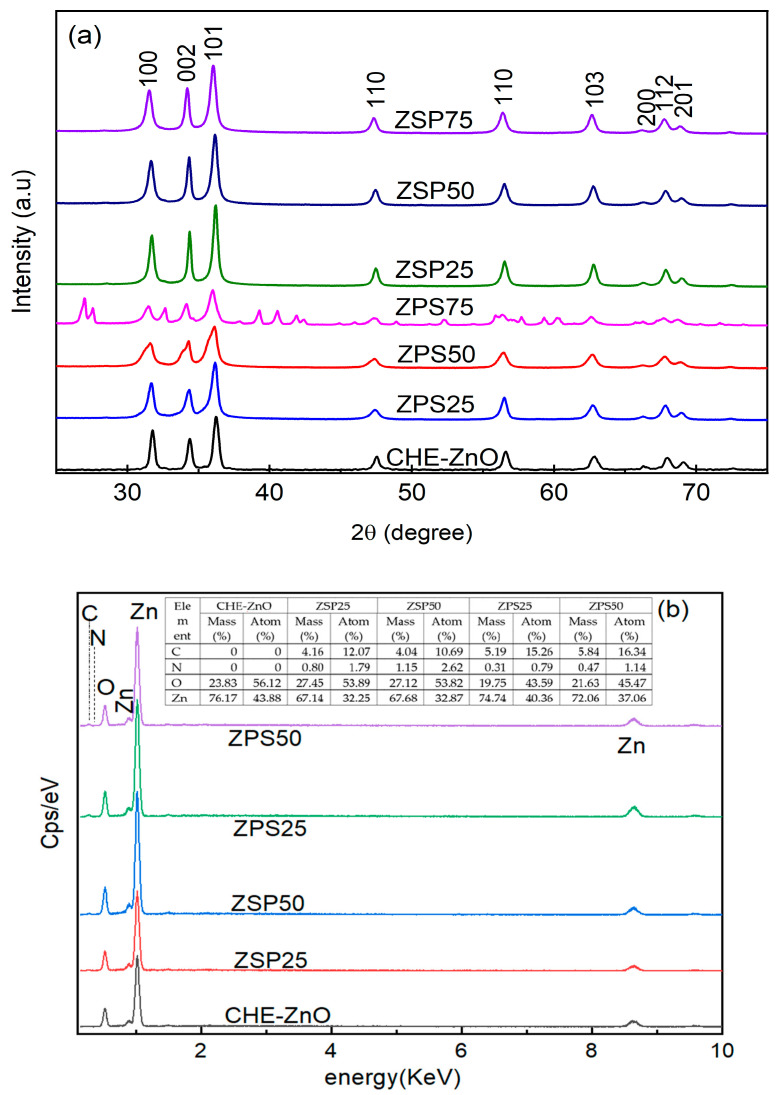
(**a**) X-ray diffraction pattern and (**b**) EDS spectra of some synthesized materials.

**Figure 2 materials-16-05457-f002:**
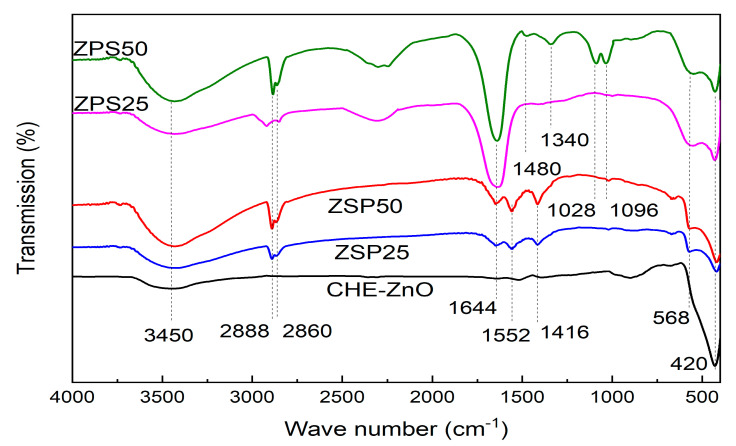
FTIR spectra of synthesized samples.

**Figure 3 materials-16-05457-f003:**
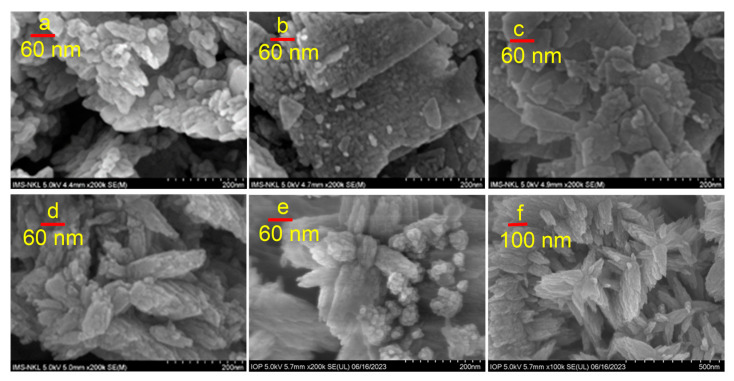
SEM images of synthesized CHE-ZnO NPs (**a**); ZSP25 (**b**); ZSP50 (**c**); ZPS25 (**d**); and ZPS50 (**e**,**f**).

**Figure 4 materials-16-05457-f004:**
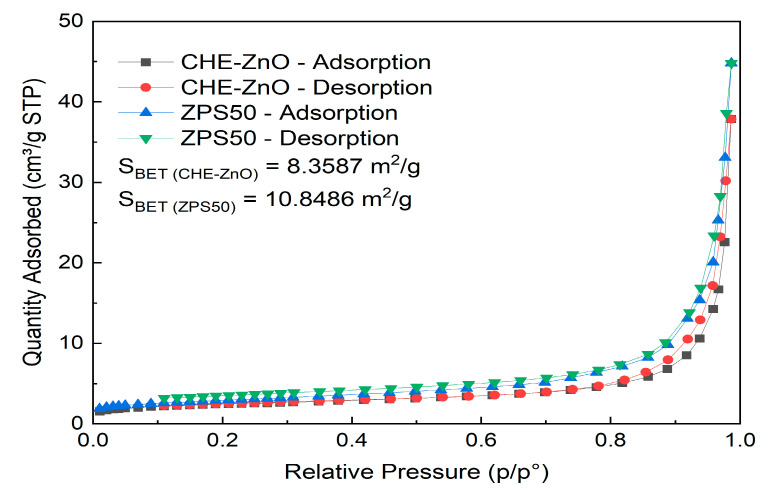
The curve of adsorption and desorption of N2 gas of CHE-ZnO and ZPS50 nanoparticles.

**Figure 5 materials-16-05457-f005:**
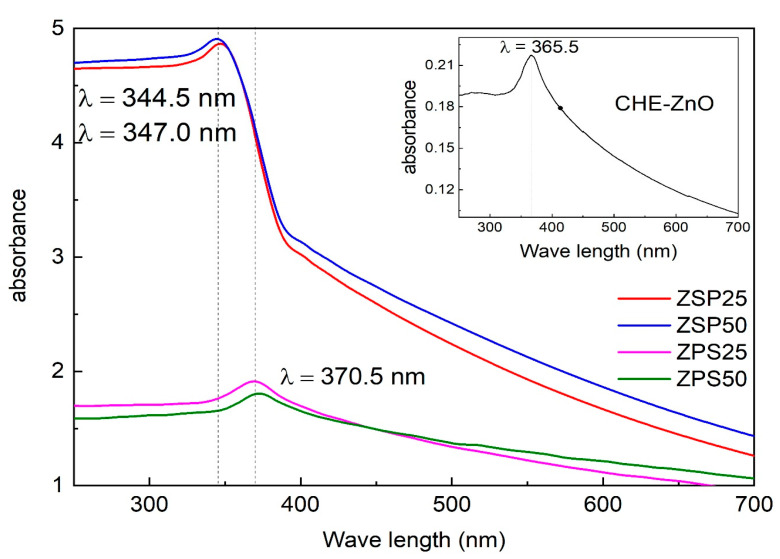
UV-Vis DRS spectra of power of synthesized samples.

**Figure 6 materials-16-05457-f006:**
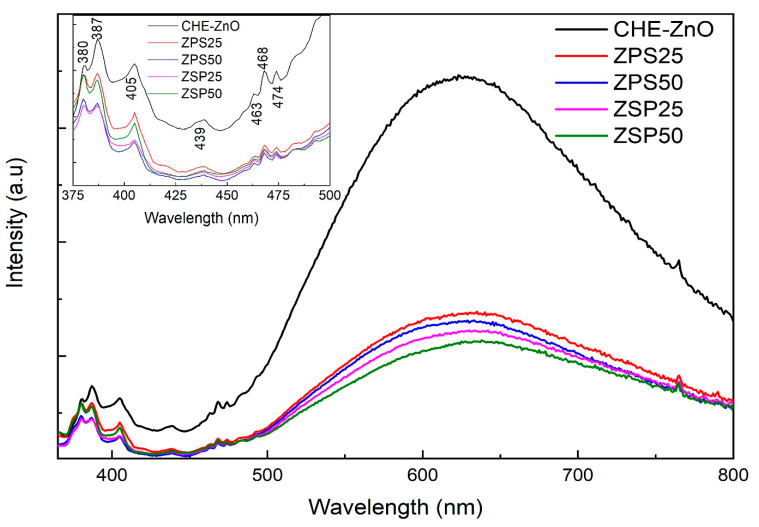
Photoluminescence spectra of the synthesized samples.

**Figure 7 materials-16-05457-f007:**
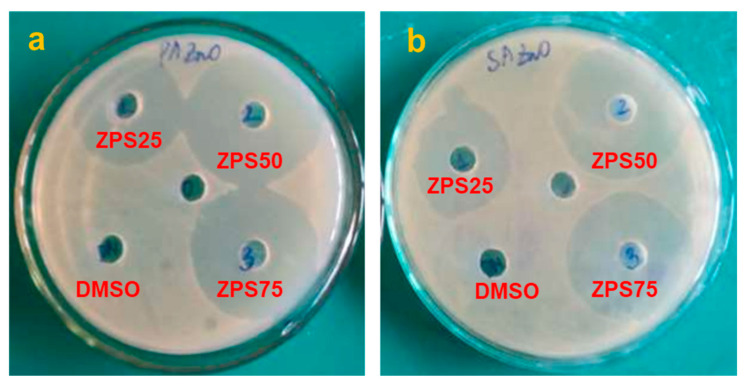
Antibacterial activity of ZPS25, ZPS50, and ZSP75 samples and DMSO solvent against (**a**) *P. aeruginosa* and (**b**) *S. aureus*.

## Data Availability

Not applicable.

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
