# Peer review of "Characterization and Bioactivity of Piper chaudocanum L. Extract-Doped ZnO Nanoparticles Biosynthesized by Co-Precipitation Method"

_materials, 2023, doi:10.3390/ma16155457_

Round 1

Reviewer 1 Report

The manuscript is not suitable for publication in this journal for the following reasons:

1- The study is traditional.

2- The manuscript needs an extensive study of nanoparticles (the behavior of nanoparticles, how to calculate their surface area,and the effect of nanoparticles).

3- The images of the SEM were in the form of islands, and therefore their efficiency in killing bacteria is not clear, the nanoparticles produced in this way are inefficient in killing bacteria.

Moderate editing of English language required.

Author Response

Dear Editor and reviewers, We would like to express our gratitude for the Editor and Reviewer’s efforts to improve the quality of our manuscript. We have tried our best to respond to all issues indicated in the review report sufficiently. In the revised version, we have highlighted the changes to our manuscript using the track changes. The answers to the questions you raised are detailed here.

Reviewer 2 Report

Dear Authors,

The  paper ”Characterization of Piper chaudocanum L. extract doped ZnO nanoparticles biosynthesized by co-precipitation method and  bioactivity” presents a biosynthesis of PLE-doped ZnO NPs, by the co-precipitation method.

1. it is worth appreciating the fact that the synthesis stages were varied, thus observing the influence of these parameters on the morphology and size of the nanoparticles.

2. usual characterization methods were used, but transmission microscopy is still missing, which would highlight much better the morphology and size of the obtained nanoparticles. also, it is necessary to reset the SEM images (figure 3) depending on the sample series and magnification. An energy -dispersive X-ray analysis  (EDX or EDS) could have quickly generated information about the chemical composition of a sample, including which elements are present, as well as their distribution and concentration.

3. the bibliography is appropriate, current.

4. some editorial corrections are needed (eg. EPL-doped ZnO NPs, 394).

Author Response

(The authors gave the same response as above.)

Reviewer 3 Report

Dear Authors, in your manuscript, the following points should be added/changed to further improve it:

1.      Title: Please explain why the authors used the phrase "Piper chaudocanum L. ex-2 tract doped ZnO nanoparticles". Is this consistent with the definition of 'doped ZnO'? What dopant in the ZnO crystal lattice is being referred to here?

2.      Introduction: I ask the authors to add in the text the phrase that zinc oxide is a "multifunctional material”.

3.      Introduction: I have a comment on the following sentence “Many physical and chemical techniques, such as sol-gel, hydrothermal, mechano-47 chemical, hydrometallurgy, and thermal decomposition, have been employed to synthe-48 size ZnO NPs [7].” I suggest adding references to significant review publications on ZnO synthesis (e.g. DOI:10.1016/j.reffit.2017.03.002; DOI: 10.3390/nano10061086).

4.      XRD Patterns: The name "Debye-Scherrer equation" is incorrect. The correct name is "Scherrer equation" (DOI: 10.1038/nnano.2011.145). The size of the crystallites is not the size of the particles unless a monocrystalline particle is obtained.

5.      Figure 2. FTIR spectra of synthesized samples: Please correct errors in the determination of absorption band values.

6.      SEM imagines prepared samples: The authors claim in their title that they obtained ZnO nanoparticles, however, SEM images contradict this. I suggest introducing the concept of ZnO nanostructure. The authors report the size of ZnO as the diameter of the crystallites, which has nothing to do with the real size of the structures obtained. Please sort out the issue of ZnO size throughout the manuscript.

7.      Antibacterial activity: I have a comment on the following sentence “ZPS50 NPs exhibited the best antibacterial 372 activity due to their smaller diameters and more modified surface by P.chaudocanum ex-373 tract.” Have the authors considered the possibility that in this case only the content of the modifier in the sample matters (purity of the sample) ?

8.      Antibacterial activity: No supplementary material was added to the manuscript.

9.      Antibacterial activity: Where are the results for sample CHE-ZnO?

Author Response

(The authors gave the same response as above.)

Reviewer 4 Report

According to my opinion, the manuscript entitled "Characterization and bioactivity of Piper chaudocanum L. extract doped ZnO nanoparticles biosynthesized by co-precipitation method"  given by Thi Thao Truong, Thi Tam Khieu, Huu Nguyen Luu, Hai Bang Truong, Van Khien Nguyen, Truong Xuan Vuong and Thi Kim Ngan Tran is appropriate for Materials. I share the Authors’ opinion that cytotoxic activity of ZPS50 NPs ZPS50 NPs, a PLE-doped ZnO NPs, could be a potential material for developing drugs which could treat human epithelial carcinoma cells and infectious illnesses. I did not find any mistakes in the manuscript. Therefore, I recommend to accept it in the current form. Additionally, I would like to add that all concluding remarks were undoubtedly confirmed by experiments.

Author Response

(The authors gave the same response as above.)

Round 2

Reviewer 2 Report

Dear Authors,

I appreciate the changes made, as a result I propose the publication of the work.